# CD44, γ-H2AX, and p-ATM Expressions in Short-Term Ex Vivo Culture of Tumour Slices Predict the Treatment Response in Patients with Oral Squamous Cell Carcinoma

**DOI:** 10.3390/ijms23020877

**Published:** 2022-01-14

**Authors:** Pierre Philouze, Arnaud Gauthier, Alexandra Lauret, Céline Malesys, Giovanna Muggiolu, Sylvie Sauvaigo, Antoine Galmiche, Philippe Ceruse, Gersende Alphonse, Anne-Sophie Wozny, Claire Rodriguez-Lafrasse

**Affiliations:** 1Laboratory of Cellular and Molecular Radiobiology, UMR CNRS5822/IP2I, Lyon-Sud Medical School, Univ Lyon 1, Lyon University, 69921 Oullins, France; pierre.philouze@chu-lyon.fr (P.P.); arnaud.gauthier@univ-lyon1.fr (A.G.); alexandra.lauret@univ-lyon1.fr (A.L.); celine.malesys@univ-lyon1.fr (C.M.); philippe.ceruse@chu-lyon.fr (P.C.); anne-sophie.wozny@univ-lyon1.fr (A.-S.W.); 2Head and Neck Department, Croix-Rousse Hospital, Hospices Civils of Lyon, 69004 Lyon, France; 3Department of Biochemistry and Molecular Biology, Lyon-Sud Hospital, Hospices Civils of Lyon, 69310 Pierre-Bénite, France; 4LXRepair, 38700 La Tronche, France; giovanna.muggiolu@lxrepair.com (G.M.); sylvie.sauvaigo@lxrepair.com (S.S.); 5UMR7516 “CHIMERE”, University of Picardie Jules Verne—Department of Biochemistry of the Amiens Hospital, 80054 Amiens, France; galmiche.Antoine@chu-amiens.fr

**Keywords:** HNSCC, γ-H2AX foci, pATM, DNA damage response, ex vivo culture, CD44, predictive biomarker

## Abstract

Squamous cell carcinoma is the most common type of head and neck cancer (HNSCC) with a disease-free survival at 3 years that does not exceed 30%. Biomarkers able to predict clinical outcomes are clearly needed. The purpose of this study was to investigate whether a short-term culture of tumour fragments irradiated ex vivo could anticipate patient responses to chemo- and/or radiotherapies. Biopsies were collected prior to treatment from a cohort of 28 patients with non-operable tumours of the oral cavity or oropharynx, and then cultured ex vivo. Short-term biopsy slice culture is a robust method that keeps cells viable for 7 days. Different biomarkers involved in the stemness status (CD44) or the DNA damage response (pATM and γ-H2AX) were investigated for their potential to predict the treatment response. A higher expression of all these markers was predictive of a poor response to treatment. This allowed the stratification of responder or non-responder patients to treatment. Moreover, the ratio for the expression of the three markers 24 h after 4 Gy irradiation versus 0 Gy was higher in responder than in non-responder patients. Finally, combining these biomarkers greatly improved their predictive potential, especially when the γ-H2AX ratio was associated with the CD44 ratio or the pATM ratio. These results encourage further evaluation of these biomarkers in a larger cohort of patients.

## 1. Introduction

Head and Neck Squamous Cell Carcinoma (HNSCC), including oral squamous cell carcinoma and oropharynx tumour, is the most common type of head and neck cancer [1]. Different therapeutic strategies are used for the non-operable stages (III–IV) of HNSCC, including induction chemotherapy (cisplatin, 5-fluorouracil (5-FU) and docetaxel) followed by radiotherapy, radiotherapy alone, or more recently the use of immune checkpoint inhibitors such as pembrolizumab alone or in combination with platinum and 5-FU [2,3]. However, tumours display heterogeneity in terms of clinical outcome and response to treatment, even among patients who are assigned with the same level of risk.

Therefore, biomarkers are needed to help clinicians choose the best treatment for personalized therapy. Although numerous alterations at the genomic [4], proteomic [5,6], and radiomic levels [7] have been proposed as predictive biomarkers of a treatment response, very few have been validated up to now [8]. Human papillomavirus (HPV) expression is broadly recognized by clinicians as a predictive biomarker in HNSCC. HPV-positive tumours respond better to radio- or chemotherapy prompting a de-escalation of treatment, whereas HPV-negative tumours show a worse response [9].

Other prognosis or predictive biomarkers seem very promising, but are not yet transferred to the clinic due to lower availability of data. This is the case for CD44, a cancer stem cell (CSC) marker involved in HNSCC tumour growth and metastasis [10,11]. Several studies have demonstrated that HNSCC tumours enriched with CD44-positive cells are associated with local recurrence after chemo- and radio-therapy [10,12,13]. These studies have all focused on the basal level of CD44 expression, but it has been shown in vitro that, depending on the radiosensitivity of cells, photon irradiation induced a CD44 enrichment leading to an increased radioresistance [14]. Understanding whether this mechanism is induced during radiotherapy would therefore allow for better treatment management. 

Exposure to both radiation and cytotoxic agents induces DNA damage, including DNA double-strand breaks (DNA-DSBs). These damages lead to activation of the DNA damage response (DDR) that controls cell-cycle checkpoints, DNA repair and apoptosis. The protein kinase ataxia-telangiectasia mutated protein (ATM) plays a central role in DDR activation [15]. When DNA-DSBs are induced, ATM is phosphorylated at its serine 1981 residue (pATM) and then induces the phosphorylation of key proteins, including histone H2AX to generate γ-H2AX [16,17]. The expression of both pATM and γ-H2AX thus plays a critical role in controlling DNA repair [18]. Indeed, their expression could be used as predictive biomarkers of patient survival. In patients with nasopharyngeal cancer receiving chemoradiotherapy or with glioblastoma multiforme, high basal ATM protein expression is associated with poor overall survival (OS) [19,20]. In breast cancer, the opposite result was obtained with low basal ATM expression associated with poor OS. As for CD44, these results were obtained only by analysing basal ATM expression. Since ATM is phosphorylated and activated in response to DNA damage, studying its expression after irradiation seems more appropriate. Concerning γ-H2AX, its quantification is only used in normal cells to predict the toxicity of different treatments [21]. Moreover, after irradiation of human HNSCC xenografted tumours, the residual γ-H2AX foci could predict the radiation response [22,23]. 

These different results encouraged us to develop an assay based on CD44 and DDR pathway quantification for clinical application to determine treatment outcomes. Tumour biopsies, collected from patients prior to treatment, were sliced, maintained in ex vivo culture, and irradiated or not. The different biomarkers were then analysed on slices by microscopy after immunostaining, in order to predict the ability of the patient to respond successfully or not to the treatment. 

## 2. Results

### 2.1. Patients

All 28 patients were diagnosed histologically with HNSCC localized in the oropharynx, the oral cavity (oral squamous cell carcinoma) or both (Table 1). Five patients received radiotherapy and 23 were treated by induction chemotherapy (cisplatin, 5-FU and, docetaxel) ± radiotherapy. According to the clinician evaluation at 12 months, the patients were classified into two groups: responders (patients who no longer had tumours at the primary site), and non-responders (patients with partial response or dead). Using this classification, 15 patients were responders and 13 were non-responders. Among the 28 patients, six were HPV-positive, and eight had a p53 mutation. No correlation was found between these markers and the response to treatment.

### 2.2. Viability of Biopsy Sections Maintained in Ex Vivo Culture

Cell proliferation was assessed by Ki67 labelling on four biopsies and the percentages of negative and positive Ki67 cells were quantified (Figure 1A,B). The labelling shows that the biopsy sections maintained in ex vivo culture represent a majority of proliferative cells even at 7 d (66.5 ± 11.4%). Apoptotic death was also evaluated using TUNEL labelling (Figure 1C,D). The percentage of apoptotic cells 30 min after ex vivo culture only reaches 3.7 ± 2.4% and slightly increases over time to reach 27.9 ± 7.7% after 7 d. These results show that the biopsies are viable for at least 7 d in ex vivo culture.

### 2.3. CD44 Expression 

CD44 expression was quantified in ex vivo irradiated or non-irradiated tumour slices, obtained from biopsy of patients prior to treatment, in order to investigate whether irradiation could improve the predictive value of this CSC marker (Figure 2). Figure 2A shows representative images of CD44 labelling for responder and non-responder patients, whereas Figure 2B displays the mean ± standard derivation (SD) of CD44 intensity in arbitrary units. Basal CD44 expression is significantly lower in responder patients (2256 ± 704) compared with non-responder ones (3832 ± 1875) (* *p* < 0.05). After a 4 Gy irradiation, the CD44 expression increases in both groups. The ratio of CD44 expression after irradiation compared to its basal level was calculated, but no significant difference was observed between the two patient groups (Figure 2C). 

Based on the basal expression level of CD44, a threshold that determines whether a patient responds to treatment was defined at 2760 using the Area Under the Receiver Operating Characteristic Curve (AUROC) statistical analysis. The corresponding curves are shown in Appendix A. Patients with a score above this cut-off point are considered non-responders to the treatment, while patients with a lower score are considered responders. The area under the curve (AUC), which can be used as a criterion to measure the marker’s discriminative ability, was also calculated (Table 2). With an AUC of 0.789, CD44 could be considered as a good predictive marker.

### 2.4. pATM and γ-H2AX Expression in Ex Vivo Irradiated and Non-Irradiated Tumour Slices

As the DDR pathways play a major role in the balance between cell survival and cell death, the evaluation of pATM expression and γ-H2AX foci as predictive biomarkers were undertaken (Figure 3 and Figure 4). Depending on the response to treatment, two different profiles of pATM can be distinguished (Figure 3). In the absence of irradiation, non-responder patients have a significantly higher percentage of pATM than responders (34.8 ± 12.4% vs. 13.1 ± 12.6% respectively (* *p* < 0.05)) (Figure 3B). After a 4 Gy irradiation, only the biopsies of the responder patients showed a significantly higher percentage of pATM (39.5 ± 15.1%) compared with the non-irradiated ones (13.1 ± 12.6%) (** *p* < 0.01). The 4 Gy/0 Gy ratio for pATM intensity was much higher for responder patients compared to non-responder ones (Figure 3C). However, even if a clear tendency was observed, this difference was not significant. The threshold that allows differentiating the two groups was set to 24% for the basal level of pATM. The AUC was determined at 0.56, which is less performant than CD44 (Table 2).

Concerning the γ-H2AX foci, Figure 4A shows representative cells at 24 h containing γ-H2AX foci in non-irradiated or irradiated cells from a responder patient. Figure 4B presents the mean number ± SD of γ-H2AX foci per cell. When the two groups are compared, it emerges that in the absence of irradiation, the mean number of γ-H2AX foci is significantly higher (** *p* < 0.01) for non-responder patients (3.2 ± 0.5) compared with responder patients (0.8 ± 0.7). After irradiation, no significant difference was observed between the residual γ-H2AX foci of the responder (2.5 ± 2.1) and non-responder patients (3.9 ± 2.3). Figure 4C shows two different profiles of the 4 Gy/0 Gy ratio, depending on whether the patients are responders or non-responders. For this marker, the γ-H2AX ratio obtained at 24 h is higher (* *p* < 0.05) for responder patients than for non-responder patients. The threshold for classifying patients has been determined at 1.4 γ-H2AX foci for non-irradiated cells. Moreover, in contrast to CD44 and pATM markers, the γ-H2AX ratio 4 Gy/0 Gy was significantly different between responders and non-responders (* *p* < 0.05). The threshold ratio was, therefore, calculated and set at 1.6.

### 2.5. Combining Markers to Improve the Prediction of Treatment Response

As the three markers, CD44, pATM, and γ-H2AX, taken independently, statistically predict the response to treatment, it was important to define if (i) adding irradiation to the basal marker level or (ii) combining the markers could improve the discrimination between responders and non-responders. ROC (receiver operating characteristic curve) analyses and AUC were then calculated (Table 2). It appears that the best biomarker is the 4 Gy/0 Gy ratio of γ-H2AX with an AUC of 0.875. The AUC of the different combinations were calculated without irradiation and the only pertinent association to improve sensitivity was CD44 and γ-H2AX. Very interestingly, when irradiation is added, the combination of ratios results in a higher AUC, especially for combinations involving the γ-H2AX ratio. 

## 3. Discussion

The objective of this clinical study was, using biopsies collected prior to treatment, to determine if the irradiation of ex vivo short-term culture of biopsy slices was suitable for the determination of predictive biomarkers of oral squamous cells carcinomas and oropharynx tumour with regard to chemo- or radio- therapy treatment. 

Short-term culture of tumour slices is an innovative way to predict the response to chemo- or radiotherapy [24,25,26,27]. First, an essential requirement for the validation of biomarkers was to guarantee that the ex vivo biopsies were viable in culture. Proliferation and apoptosis induction were evaluated, and high cell survival was observed for up to 7 d. Short-term culture of tumour slices has been used previously for the evaluation of individual sensitivity to different chemotherapies [27,28] and have revealed the heterogeneous sensitivity of patients with HNSCC to cetuximab [29]. Here, in contrast to others, who focused on a single treatment (radiotherapy or chemotherapy), we showed that ex vivo culture could be used for the determination of biomarkers predictive of the response to both chemo- and radio- therapy in patients with oral squamous cell carcinoma or oropharynx tumour.

CSCs have been shown to induce tumour progression and resistance to treatment [30,31]. Therefore, different stem cell markers, such as CD44, have already been studied in solid tumours [32]. While most of those studies showed that high CD44 expression is associated with advanced disease, metastasis, invasion and, lower OS [30,33], others have reported that a low CD44 expression correlates with more aggressive tumours [34] or described a lack of correlation between survival and CD44 expression [35]. In this study, a significant difference in basal CD44 expression between the responder and non-responder patients was shown, confirming the observation that high expression of CD44 is associated with poor prognosis. 

We then investigated the predictive value of the expression of proteins involved in the DDR. Our results highlighted that high pATM expression is correlated with a poor response to chemo- or radio- therapy treatments. Moreover, whereas the percentage of pATM remains high 24 h after a 4 Gy irradiation, in the responder patients, it returns almost to the basal value for the non-responders. To our knowledge, this is the first time that pATM has been assessed by immunohistochemistry before and after irradiation using biopsies from patients with HNSCC. Only one study has investigated the expression of pATM in correlation with radiation response as a prognosis biomarker in supraglottic tumours [36]. In cervical carcinoma, high levels of pATM were also associated with poor locoregional disease-free survival [37]. 

Quantification of residual DNA-DSBs was also investigated to define the number of γ-H2AX foci that could predict patient outcomes. The majority of published works that focused on the role of γ-H2AX foci as a biomarker to predict tumour radiosensitivity [38,39] were conducted in vitro or in HNSCC tumour xenograft models and not in a clinical study. [40]. The results obtained here show that for the responder patients, the mean number of γ-H2AX foci per cell without irradiation or the 4 Gy/0 Gy ratio are both lower for the non-responder patients. By irradiating the biopsy sections maintained in ex vivo culture and calculating the 4 Gy/0 Gy ratio, the AUC, and thus the sensitivity and the specificity of the test, were increased. Rassamegevanon et al. also found that γ-H2AX foci from ex vivo cultured tumour biopsies could reflect and predict the radiation response of the corresponding tumours irradiated in viivo [41]. Finally, the combination of the γ-H2AX foci and pATM ratios, two biomarkers involved in DDR, significantly improves the prediction of response.

## 4. Materials and Methods

### 4.1. Patients

Twenty-eight patients, who had been referred to the Head and Neck Department of Croix-Rousse Hospital (Lyon, France) with a histologically proven HNSCC from the oral cavity or the oropharynx were recruited between May 2016 and November 2018. All patients had non-operable tumours. They were treated by induction chemotherapy (cisplatin, 5-FU and, docetaxel) followed by adjuvant radiotherapy or radiotherapy. Patients were included for a period of 2 years and were monitored for 12 months after the end of their treatment. This study was performed under the clinical trial ChemRadAssay (DNA Repair enzyme signature associated with response to chemo- and radiotherapy in Head and Neck cancer, NCT02714920), as an ancillary study. Written consent was obtained from each patient. No sample was obtained from minors (<18 years old) or physically and/or mentally impaired patients unable to understand and give their consent to the use of their samples.

### 4.2. Tumour Samples

A biopsy was obtained under local anaesthesia before the beginning of patient’s treatment and was immediately immersed in Dulbecco’s modified Eagle’s medium (DMEM, Invitrogen, Carlsbad, CA, USA) supplemented with 10% foetal bovine serum (FBS, Dutscher, Brumath, France). A summary diagram of the main steps of sample processing is shown in Figure 5.

### 4.3. Ex Vivo Culture of Tumour Biopsy

The biopsy was embedded in 1.5% agarose diluted in DMEM. The block obtained was mounted on the specimen holder of a Vibratome VT1200S (Leica, Nanterre, France) and immersed in 150 mL of DMEM containing 10% FBS, 1% penicillin/streptomycin (Invitrogen). Automated slicing was then performed. Tumour slices of 300 μm were collected and placed in 6-well plates containing a mix of DMEM and 10% FBS. The slices were cultivated at 37 °C, 5% CO_2,_ under soft rotation. 

### 4.4. Tumour Slice Irradiation

For each patient, one ex vivo tumour slice was irradiated at 4 Gy at an energy of 250 kV with a dose-rate of 2 Gy·min^−1^ (X-Rad320 irradiator, Precision X-ray, North Branford, CT, USA) [24,25]. Tumour slices were immediately replaced in the incubator for 24 h. As a control, a non-irradiated tumour slice was analysed in parallel. 

### 4.5. Immunohistochemical (IHC) Analyses

At different times, the tumour slices were fixed 1 h in 4% formol, then washed with phosphate-buffered saline (PBS) (Invitrogen), and embedded in paraffin. Sections of 5 μm thickness were finally cut with a Microtome (Shandon Finesse E/ME, Thermo Scientific, Waltham, MA, USA) and mounted on SuperFrost Gold Slides (Fisher Scientific, Illkirch, France).

For each biopsy, a slice was stained with haematoxylin-eosin, and analysed to select non-necrotic/non-fibrotic areas with a high density of tumour cells. The defined regions were memorized using the Metafer system (MetaSystems, Altlussheim, Germany). Then, the stained slides were placed on the Microscope Axio Imager Z2 (Zeiss, Marly-Le-Roi, France), and only the previously defined areas, i.e., the tumoral zone, were analysed.

#### 4.5.1. Cell Viability

Tumour sections (*n* = 4) were prepared for different time-point analyses (30 min to 7 d) and stained using the EnVision FLEX Mini-Kit, Low pH (Dako, Agilent, Santa-Clara, CA, USA) according to the manufacturer’s specifications. Ki67 mouse monoclonal antibody was used at 1:50 (clone-MIB1, Dako). The Ki67-positive cells were counted in 200 random fields (20×). Depending on the intensity of the staining, two classes were defined: “−” for none or little staining, “+” for medium and strong staining.

#### 4.5.2. Quantification of CD44 and pATM

Immunostainings of CD44 and pATM were performed according to the protocol used for Ki67. However, tissue sections were incubated with 1:100 CD44 rabbit monoclonal antibody (E7K2Y, Cell Signaling, Danvers, MA, USA) in EnVision FLEX antibody diluent overnight at 4 °C or with 1:70 pATM rabbit monoclonal antibody (phospho S1981, Abcam, Cambridge, UK) 2 h at 37 °C in TBS (Tris Buffered Saline), 1% BSA (Bovine Serum Albumin, Sigma (Merck, Darmstadt, Germany)). The CD44 intensity was quantified as the intensity of brown staining in 200 random fields (20×). For pATM, depending on the intensity of the cell staining, two groups were defined: “−” for none or little staining and “+” for medium and strong staining.

### 4.6. Immunofluorescence (IF) Analysis

#### 4.6.1. Quantification of Apoptotic Cells

Terminal deoxynucleotidyl transferase dUTP Nick End Labelling (TUNEL) was carried out with the Dead End Fluorometric TUNEL System (Promega, Madison, WI, USA) on paraffin-embedded sections according to the manufacturer’s instructions. The fluorescent cells were quantified for 200 fields and the percentage of TUNEL positive cells was established (20×).

#### 4.6.2. Quantification of γ-H2AX Foci

Paraffin-embedded sections were successively rinsed in Ottix-Plus (MicromMicrotec, Brignais, France), OttixShapper (MicromMicrotec), and water. The slides were then incubated in unmasking buffer (100 mM TrisBase, 10 mM EDTA, 0.05% Tween 20, pH 9) for 30 min at 99 °C, before being permeabilized in PBS, 0.2% Triton X-100. After washing (PBS, 0.1% Tween 20, 0.05% TritonX-100), and blocking (PBS, 0.2% milk, 5% FBS, 0.1% Triton X-100) for 10 min, the slides were incubated with a 1:50 γ-H2AX mouse monoclonal antibody (Millipore, Burlington, VT, USA) overnight at 4 °C, and then with a 1:500 anti-mouse IgG-Alexa Fluor 488 at 37 °C for 2 h (Thermo Fischer Scientific, Waltham, MA, USA). After 3 washes in PBS, 0.1% Tween 20, and two in PBS alone, slides were incubated in 1 µg·mL^−1^ DAPI (Merck) for 10 min and finally mounted with Fluoromount^®^ (Merck). Two slides and at least 400 nuclei were analysed for each condition. The results were expressed as the mean number of foci per nucleus.

### 4.7. Statistical Analysis

The Student’s *t*-test was used and the minimum level of significance was set at *p* < 0.05 (* *p* < 0.05, ** *p* < 0.01, and *** *p* < 0.005, **** *p* < 0.001) (GraphPad Prism 8.4.2, GraphPad Software, San Diego, CA, USA). The ROC curves and the optimal cut-off values for CD44, pATM expression and γ-H2AX foci were calculated using GraphPad Prism 8.4.2 and the AUROC using EasyRoc 1.3.1 [42]).

## 5. Conclusions

Using short-term ex vivo culture of tumour slices, obtained from biopsies collected from patients with oral squamous cell carcinoma or oropharynx tumour before the beginning of any treatment, we were able to establish a relationship between the response to treatment and the expression of CD44, pATM, and γ-H2AX. Taken independently, each marker could determine the patient status as responder or non-responder. Furthermore, adding irradiation of ex vivo biopsies improved the discrimination between both groups. To increase the specificity and sensitivity of the prediction, a combination of these markers, particularly those involved in DDR is recommended. A confirmation of these very interesting results with a larger external cohort is now needed.

## Figures and Tables

**Figure 1 ijms-23-00877-f001:**
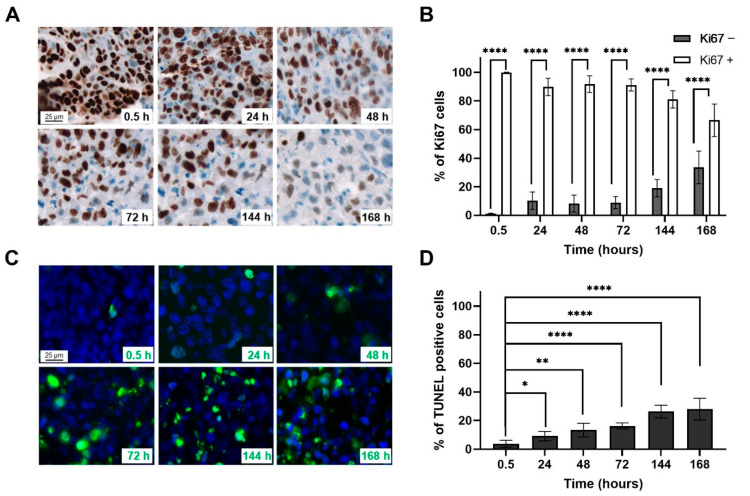
(**A**)**.** Representative microscopic acquisitions (20×) showing tumour cell proliferation assessed by Ki67 staining in tumour slices after different culture times. (**B**) Average percentage ± Standard Deviation (SD) of Ki67 tumour nuclei following categories based on the nuclear intensity: grade − (none and weak brown staining); grade + (moderate and strong brown staining). (**C**) Representative microscopic acquisitions (20×) showing apoptotic tumour cells assessed by TUNEL staining in tumour slices after different culture times. (**D**) Average percentage ± SD of apoptotic tumour cell nuclei. A minimum of 200 fields per slide were analysed per condition. Student’s *t*-test was used for statistical analyses (* *p* < 0.05, ** *p* < 0.01, **** *p* < 0.0001).

**Figure 2 ijms-23-00877-f002:**
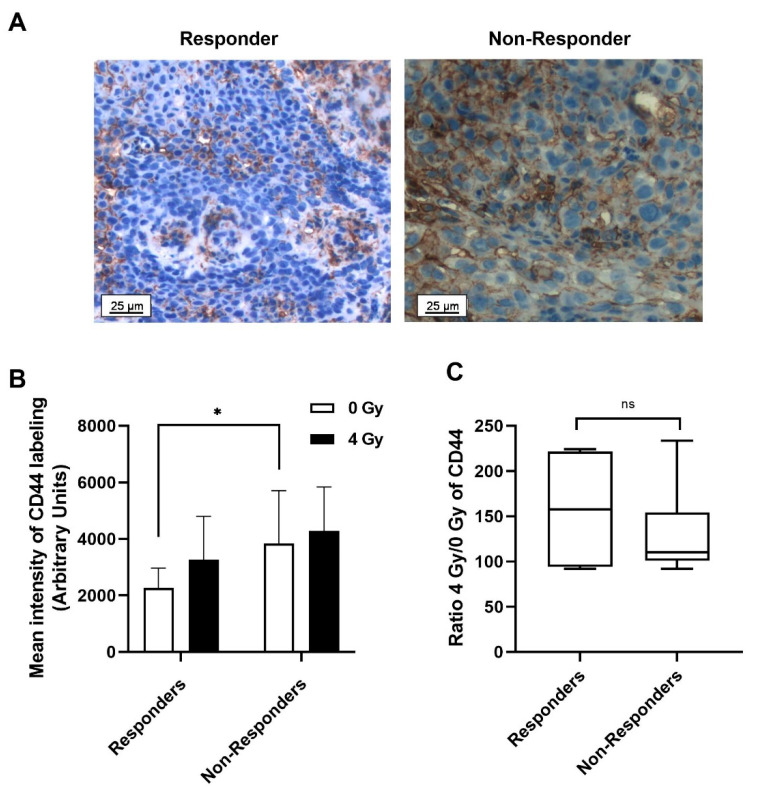
(**A**) Representative immunohistochemical (IHC) staining of CD44 expression of responder and non-responder patients in the biopsy sections maintained in ex vivo culture. (**B**) Mean ± SD of CD44 labelling intensities, quantified using Metafer software (Metasystems, Altlussheim, Germany) for 15 responder and 13 non-responder patients. (**C**) Ratio of CD44 expression after 4 Gy irradiation compared with 0 Gy for responder and non-responder patients. Two slides and a minimum of 200 fields per slide were analysed per condition. Student’s *t*-test was used for statistical analyses (^ns^ *p* > 0.05, * *p* < 0.05).

**Figure 3 ijms-23-00877-f003:**
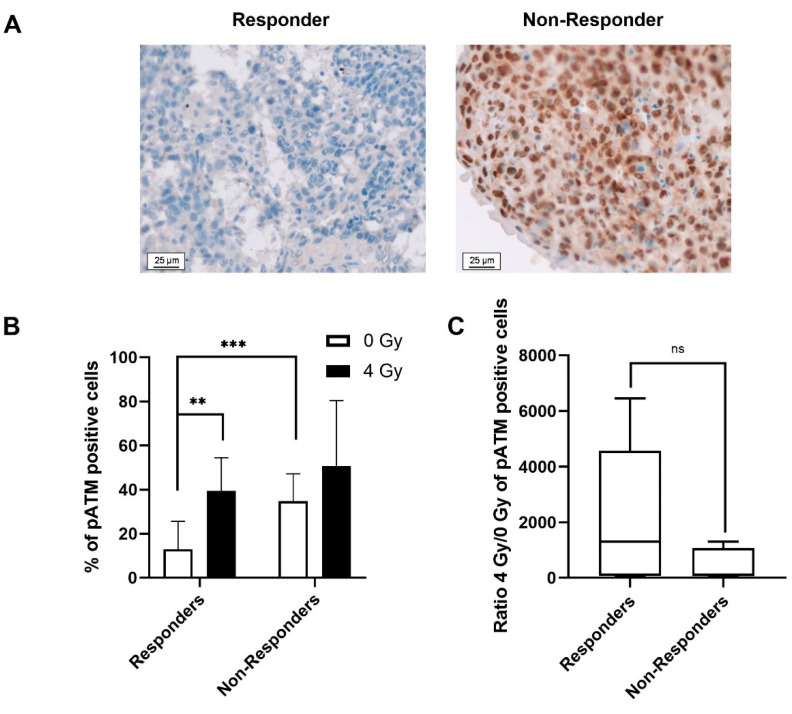
(**A**) Representative microscopic acquisition showing pATM in non-irradiated tumour slices for responders and non-responders. (**B**) Average percentage ± SD of pATM positive cells 24 h after 0 or 4 Gy irradiation for responder and non-responder patients. (**C**) Ratio of pATM expression after 4 Gy irradiation compared with 0 Gy for responder and non-responder patients. Two slides and a minimum of 200 nuclei per slide were analysed per condition. Student’s *t*-test was used for statistical analyses (^ns^ *p* > 0.05, ** *p* < 0.01, *** *p* < 0.001).

**Figure 4 ijms-23-00877-f004:**
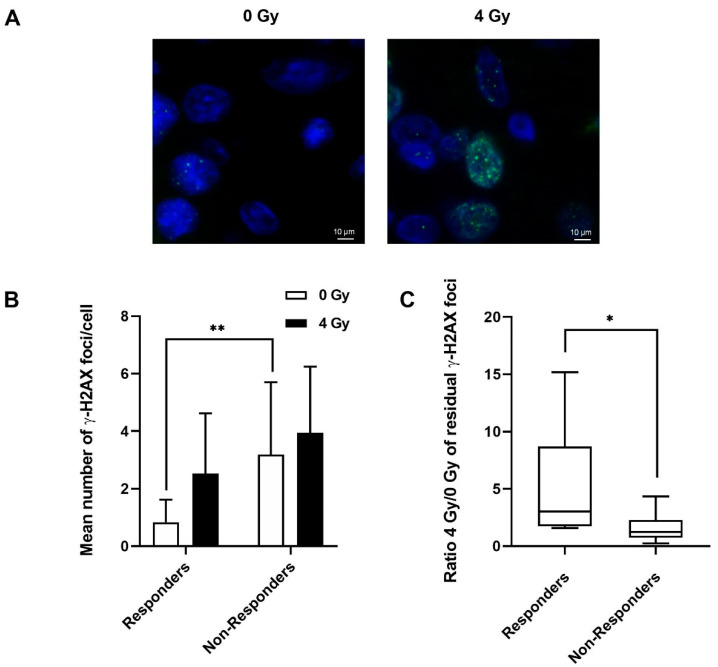
(**A**) Representative microscopic acquisition showing residual γ-H2AX foci in tumour slices of responder patients 24 h after 0 Gy or 4 Gy irradiation. (**B**) Mean number of γ-H2AX foci ± SD, 24 h after 0 Gy or 4 Gy irradiation for responder and non-responder patients. (**C**) Ratio of residual γ-H2AX foci after 4 Gy irradiation compared with 0 Gy for responder and non-responder patients. Two slides and a minimum of 400 nuclei per slide were analysed per condition. Student’s *t*-test was used for statistical analyses (* *p* < 0.05, ** *p* < 0.01).

**Figure 5 ijms-23-00877-f005:**
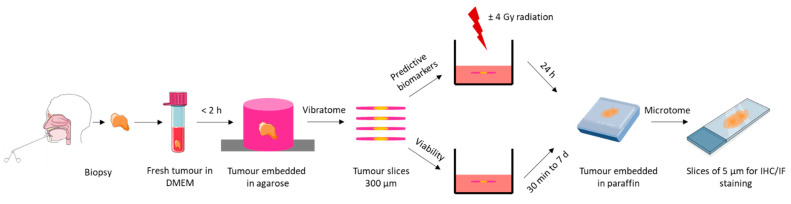
Chronological diagram of biopsy processing.

**Table 1 ijms-23-00877-t001:** Clinico-pathological characteristics of patients.

Characteristics	Value *n* (%)
**N**	28
**Age (y)**	61.8 ± 8.4
**Gender**	
Female	6 (21.4%)
Male	22 (78.6%)
**Alcohol**	
Yes	15 (53.6%)
No	11 (39.3%)
Unknown	2 (7.1%)
**Smoking**	
Yes	23 (82.1%)
No	4 (14.3%)
Unknown	1 (3.6%)
**Localisation**	
Oral Cavity	23 (82.2%)
Oropharynx	3 (10.7%)
Oropharynx/Oral Cavity	2 (7.1%)
**T-Stage**	
T1	0 (0%)
T2	7 (25.0%)
T3	9 (32.1%)
T4	12 (42.9%)
**N-Stage**	
N0	4 (14.3%)
N1	2 (7.1%)
N2	20 (71.5%)
N3	2 (7.1%)
**HPV**	
Positive	6 (21.4%)
Negative	22 (78.6%)
**P53**	
Mutated	8 (28.6%)
Wild-Type	20 (71.4%)
**Treatment**	
Radiotherapy	5 (17.9%)
Chemotherapy	23 (82.1%)
**Response to treatment at 12 months**	
Responders-Total	15 (53.6%)
Non Responders-Total Non Responders-Partial response Non Responders-Death	13 (46.4%) 8 (28.6%) 5 (17.8%)

**Table 2 ijms-23-00877-t002:** Receiver Operating Characteristic (ROC) analyses for the different markers at basal levels (0 Gy) or after 4 Gy irradiation: 4 Gy/0 Gy ratio. Analyses were performed for all biomarkers individually or in combination.

Markers	AUC *	95% Confidence Intervals
CD44	0.789	[0.569, 1.000]
pATM	0.557	[0.238, 0.876]
γ-H2AX	0.758	[0.535, 0.981]
CD44 ratio	0.500	[0.046, 0.954]
pATM ratio	0.680	[0.284, 1.000]
γ-H2AX ratio	0.875	[0.688, 1.000]
γ-H2AX, CD44	0.844	[0.615, 1.000]
γ-H2AX, pATM	0.783	[0.542, 1.000]
CD44, pATM	0.694	[0.395, 0.992]
γ-H2AX ratio, CD44 ratio	0.917	[0.723, 1.000]
γ-H2AX ratio, pATM ratio	1.000	[1.000, 1.000]
CD44 ratio, pATM ratio	0.667	[0.013, 1.000]

* AUC: Area Under the Curve.

## Data Availability

The datasets used and analysed during in the current study are available from the corresponding author on reasonable request.

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
