# Peer review of "CD44, γ-H2AX, and p-ATM Expressions in Short-Term Ex Vivo Culture of Tumour Slices Predict the Treatment Response in Patients with Oral Squamous Cell Carcinoma"

_ijms, 2022, doi:10.3390/ijms23020877_

Round 1

Reviewer 1 Report

The manuscript entitled “Combination of biomarkers from ex vivo cultured biopsies is prognostic of the treatment response in patients with oral squamous cell carcinoma” by Philouze et al. focuses on the investigation of predictive biomarker roles of CD44 and DNA damage response (DDR) markers, pATM and γ-H2AX, measured in the short-term ex vivo culture of biopsy slices obtained from patients with oral squamous cell carcinoma. The study is scientifically sound and can be of interest to the journal audience. However, there are some concerns and recommendations:

Major concerns:

  1. This study used biopsies from 28 patients among which 23 were treated by chemotherapy and 5 were treated by radiotherapy. Therefore, levels of biomarker expression were not measured before the treatment and it is difficult to judge the predictive value of the biomarkers for the treatment response.
  2. The authors studied the effect of 4Gy irradiation on the expression of CD44, pATM, and γ-H2AX. However, the purpose of 4Gy usage in the Introduction (genotoxic or other) and its definite effects in the Results section should be explained. For example, in section 2.3, lines 131-124, and section 2.4, lines 147-150 - how to explain these effects of 4Gy?

Minor concerns:

  1. Biomarkers used to assess treatment response should be designated as predictive, but not prognostic – see doi: 1016/j.ejca.2008.03.006 and doi: 10.1016/j.hoc.2018.12.005. This should be corrected elsewhere in the manuscript. Prognostic biomarkers provide information about an overall disease outcome regardless of therapy, while predictive biomarkers are used to gain information about the efficacy and clinical benefit from surgical intervention and systemic treatment.
  2. It is better to differentiate responders and non-responders according to the type of treatment – chemotherapy, and radiotherapy.
  3. Why the authors did not use different doses (0, 1, 2, 3, and 4) of Gy? There can be dose-dependent effects of Gy/ Which dose-rate and photon energy?
  4. In the Materials and methods section, the authors mentioned the ClinicalTrials.gov identifier (NCT02714920). Does this mean that this study was performed under this clinical trial? If so, this should be discussed.

Reviewer 2 Report

Philouze and Gauthier et al. have explored the potential of using ex vivo cultured biopsies from head and neck squamous cell carcinoma (HNSCC) patients for therapeutic prognosis. For this purpose, they have devised an elegant pipeline for immunohistochemical and immunocytochemical screening of explanted HNSCC patient tumor sections following irradiation with X-rays. Based on this approach, they have discovered that the cancer stemness marker CD44 as well as the DNA damage response markers phospho-ATM(Ser1981) and γ-H2AX positively correlate with the favorable response of patients to standard radiotherapy and/or chemotherapy regimens. The authors have concluded that the combination of the ratio of both the γ-H2AX and phospho-ATM(Ser1981) staining intensity after and prior to the ex vivo irradiation was the most predictive association. The manuscript is extraordinarily well prepared and the conclusions are fully supported by the acquired data. As such, work by Philouze and Gauthier et al. has significantly pushed forward our ability to clinically diagnose HNSCC. Please see the minor points below.

1) Please provide an example of how was the area under the curve analysis presented in Table 2 performed in a new figure panel or a new figure. This could also be a new supplementary figure.

2) Please incorporate Supplementary Figure 1 as Figure 5 into the manuscript.

3) The term "oral squamous cell carcinoma" appears only in the main title but nowhere in the text. Please fix. 

4) It is not clear whether "Gersende ALPHONSE" is a corresponding author as the asterisk seems to be missing?

5) It is not clear what is the difference between "Univ Lyon" and "Lyon 1 University" as part of number 1 affiliation? Please provide unambiguous and full name for The University of Lyon.

6) Please translate "Hospices Civils de Lyon" (lines 12, 14), "Cancéropôle" (line 357), "Conseil Régional" (line 358), "Conseil Général du" (line 358), "Ligue de" (line 358), "Université de" (line 359), "Investissements d'Avenir" (line 359) into English language.

7) Please change "Lyon France" to "Lyon, France" (line 12).

8) Please replace "UMR7516 “CHIMERE”, University of Picardie Jules Verne, Department of Biochemistry" with "Department of Biochemistry, UMR7516 “CHIMERE”, University of Picardie Jules Verne" (line 17).

9) Please change "head-and-neck" to "head and neck" (lines 26, 46).

10) Please replace "then clearly" with "clearly" (line 28).

11) Please change "ex vivo, could" to "ex vivo could" (line 29).

12) Please replace "days" with "d" (lines 32, 101, 103, 104, 208, 296).

13) Please change "with" to "with the" (line 32).

14) Please replace "prognosis" with "prognostic" (line 34).

15) Please change "differentiation" to "stratification" (line 35).

16) Please change "24h" to "24 h" (lines 36, 158, 162, 170, 178 2x, 225, 282).

17) Please replace "4Gy" with "4 Gy" (lines 36, 121, 136, 147, 158, 159, 178 2x, 179, 195, 225, 280).

18) Please change "0Gy" to "0 Gy" (lines 37, 136, 159, 178 2x, 179, 195).

19) Please replace "improves" with "improved" (line 38).

20) Please change "Damage Response" to "damage response" (line 41).

21) Please replace "Head-and-Neck" with "Head and neck" (line 45).

22) Please change "[5-FU]" to "(5-FU)," (line 47).

23) Please replace "assigned to" with "assigned with" (line 51).

24) Please change "due to less availability of data, are not yet transferred to the clinic" to "are not yet transferred to the clinic due to the lack of available data" (line 60).

25) Please replace "[10,12,13] but" with "[10,12,13], but" (line 64).

26) Please change "and" to "and," (lines 68, 89, 217, 252, 353, 355).

27) Please change "ataxia-telangiectasia-mutated" to "ataxia-telangiectasia mutated" (line 69).

28) "The expression of both pATM and γ-H2AX thus plays a critical role in controlling DNA repair [17] and their expression could be used as prognosis biomarkers of patient survival. In patients with nasopharyngeal cancer receiving chemoradiotherapy or with glioblastoma multiforme, high basal ATM protein expression is associated with poor overall survival (OS)" (line 72) is too long. Please split into two sentences.

29) Please replace "in vivo after" with "after" (line 79).

30) Please change "tumoral" to "tumour" (line 83).

31) Please provide more space between line 95 and Table 1.

32) From Table 1 is not clear how many non-responder patients died?

33) Please replace "No Responders" with "Non-responders" in Table 1.

34) Please change "Fig.1A" to "Fig. 1A" (line 99).

35) Please replace "present" with "represent" (line 100).

36) Please replace "6.5±11.4%" with "6.5 ± 11.4%" (line 101).

37) Please change "Fig.1C" to "Fig. 1C" (line 102).

38) Please replace "30min" with "30 min" (lines 102, 296, 324).

39) Please change "3.7±2.4%" to "3.7 ± 2.4% (line 103).

40) Please replace "27.9±7.7%" with "27.9 ± 7.7%" (line 103).

41) Please indicate statistical significance using asterisk(s) in Figures 1B,D, 2C, 3C.

42) Please define abbreviation for "SD" (line 108), "ROC" (line 188), "CI" (Table 2), "IHC" (line 354), "IF" (line 354).

43) Please change "±S D" to "±SD" (line 111).

44) Please replace "(Fig.2). Fig.2A" with "(Fig. 2). Fig. 2A" (line 118).

45) Please change "Fig.2B displays the means" to "Fig. 2B displays the mean" (line 119).

46) Please replace "units (AI)" with "units" (line 120).

47) Please replace "2256±704" with "2256 ± 704" (line 121).

48) Please change "(3832±1875) (P<0.05)" to "(3832 ± 1875) (P < 0.05)" (line 121).

49) Please replace "Fig.2C" with "Fig. 2C" (line 124).

50) Please change "AUROC" to "area under the receiver-operating characteristic curve (AUROC)" (line 126) and "area under the receiver-operating characteristic curve (AUROC)" to "AUROC" (line 339).

51) Please replace "AUC=0.789" with "AUC = 0.789" (line 130).

52) Please increase size of the scale bar "25 μm" in both panels of Figure 2A.

53) Please change "(Arbitrary Unit)" to "(Arbitrary Units)" in the y-axis title of Figure 2B.

54) Please change "Mean±SD" to "Mean ± SD" (line 135).

55) Please replace "*P<0.05" with "* P < 0.05" (line 138).

56) Please change "Figs" to "Figs." (line 144).

57) Please replace "Fig.3" with "Fig. 3" (line 145).

58) Please change "34.8±12.4% vs 13.1±12.6% respectively (P<0.005)" to "34.8 ± 12.4% vs 13.1 ± 12.6%, respectively, P < 0.005" (line 146).

59) Please replace "Fig.3B" with "Fig. 3B" (line 147).

60) Please change "39.5±15.1 %" to "39.5 ± 15.1%" (line 148).

61) Please replace "(13.1±12.6%) (P<0.01)" with "(13.1 ± 12.6%) (P < 0.01)" (line 149).

62) Please change "4Gy/0Gy" to "4 Gy/0 Gy" (lines 149, 169, 189, 195, 238).

63) Please change "Fig.3C" to "Fig. 3C" (line 151).

64) From "Percentage±SD of pATM positive cells 24h after 0 or 4Gy irradiation for responder and non-responder patients" (line 158) is not clear whether the percentages were average?

65) Please replace "Percentage±SD" with "Percentage ± SD" (line 158).

66) Please change "*P<0.05; **P<0.01;***P<0.001" to "* P < 0.05, ** P < 0.01, *** P < 0.001" (line 161).

67) Please provide more space between lines 161 and 162.

68) Please replace "Fig.4A" with "Fig. 4A" (line 162).

69) Please change "Fig.4B" to "Fig. 4B" (line 163).

70) Please change "number±SD" to "number ± SD" (line 164).

71) Please replace "P<0.01" with "P < 0.01" (line 166).

72) Please change "3.2±.5" to "3.2 ± 0.5" (line 166).

73) Please replace "0.8±0.7" with "0.8 ± 0.7" (line 166).

74) Please change "(2.5±2.1) and non-responder patients (3.9±2.3). Fig.4C" to "(2.5 ± 2.1) and non-responder patients (3.9 ± 2.3). Fig. 4C" (line 168).

75) Please replace "P<0.05" with "P < 0.05" (line 170).

76) It is not clear what the authors refer to by "both were statistically significant" in "The threshold for classifying patients has been determined at 1.4 γ-H2AX foci per cell without irradiation and at 1.6 for the ratio, because both were statistically significant" (line 171)?

77) Please change "foci±SD" to "foci ± SD" (line 178).

78) Please replace "*P<0.05, **P<0.01" with "* P < 0.05, ** P < 0.01" (line 181).

79) It is not clear why the combination of ratio (γ-H2AX) and non-ratio (CD44) parameter "γ-H2AX Ratio, CD44" is suggested as a prognostic marker in Table 2? 

80) Please change "Ratio" to "ratio" in Table 2 10x.

81) It is not clear what the authors mean by "unique treatment" in "Here, in contrast to others who focused on a unique treatment, we showed that ex-vivo culture could be used for the determination of biomarkers prognosis of the response to both chemo- and radiotherapy in patients with HNSCC" (line 210)?

82) Please replace "others" with "others," (line 210).

83) Please change "biomarkers" to "biomarker" (line 212).

84) It is not clear what the authors mean by "both treatments" in "Our results highlighted that, before any treatment, high pATM expression is correlated with a poor response to both treatments" (line 224)?

85) Please replace "that, before any treatment, high" with "that high" (line 224).

86) Please change "24h after a 4Gy irradiation the percentage of pATM remains high" to "the percentage of pATM remains high 24h after a 4Gy irradiation" (line 225).

87) Please replace "irradiation" with "irradiation," (line 226).

88) Please change "radiation-response" to "radiation response" (line 230).

89) "The majority of studies that focused on the role of γ-H2AX foci as a biomarker to predict tumour radiosensitivity [37,38] have not been performed in a clinical study but were conducted in vitro or in HNSCC tumour xenograft models" (line 234) is not semantically correct because it is hard to conceive that a "study" could be "performed in a clinical study".

90) It is not clear what the authors mean by "Adding irradiation to the ex-vivo culture" in "Adding irradiation to the ex-vivo culture allows to increase the AUC, and therefore the sensitivity of the test" (line 239)?

91) Please replace "in-vivo irradiated tumour" with "tumours irradiated in vivo" (line 242).

92) Please change "Head-and-Neck" to "Head and Neck" (line 248).

93) Please replace "followed-up" with "followed" (line 253).

94) Please replace "minor" with "minors" (line 257).

95) Please change "Fig.1" to "Fig. 1" (line 265).

96) Please replace "tumor" with "tumour" and "Tumor" with "Tumour" 2x in Supplementary Figure 1.

97) Please replace "150mL" with "150 mL" (line 274).

98) It is not exactly clear what the authors refer to as "DMEM-10% FBS" in "Tumor slices of 300μm were collected and placed in 6-well plates containing DMEM-10% FBS" (line 275).

99) Please change "Tumor" to "Tumour" (line 275).

100) Please replace "300μm" with "300 μm" (line 275).

101) Please change "37°C" to "37 °C" (lines 276, 307, 328).

102) Please replace "X-RAD320 Irradiator" with "X-Rad320 irradiator" (line 280).

103) Please change "1h" to "1 h" (line 285).

104) Please specify the microtome used in "Sections of 5μm thickness were finally cut with a Microtome (Leica) and mounted on SuperFrost Gold Slides (Fisher Scientific, Illkirch, France)" (line 286).

105) Please replace "5μm" with "5 μm" (line 286).

106) Please change "n=4" to "n = 4" (line 296).

107) From "However, tissue sections were incubated overnight at 4°C with a 1:100 CD44 rabbit monoclonal antibody (E7K2Y,Cell Signaling, Danvers, MA, USA) in EnVision FLEX Antibody Diluent or 2h at 37°C with 1:70 pATM (phospho S1981, Abcam, Cambridge, UK)" (line 305) is not clear whether tissue sections were incubated with the pATM (phospho S1981, Abcam, Cambridge, UK) antibody in EnVision FLEX Antibody Diluent?

108) Please replace "overnight at 4°C with a 1:100 CD44 rabbit monoclonal antibody (E7K2Y,Cell Signaling, Danvers, MA, USA) in EnVision FLEX Antibody Diluent or 2h at 37°C with 1:70 pATM (phospho S1981, Abcam, Cambridge, UK)" with "with 1:100 CD44 rabbit monoclonal antibody (E7K2Y, Cell Signaling, Danvers, MA, USA) in EnVision FLEX antibody diluent overnight at 4 °C or with 1:70 pATM (phospho S1981, Abcam, Cambridge, UK) 2 h at 37 °C" (line 305).

109) It is not exactly clear what the authors mean by "PBS0.2%-TritonX-100" in "The slides were then immersed in unmasking buffer (TrisBase-100mM, EDTA-10mM, Tween20-0.05%; pH9) and incubated 30min at 99°C before being permeabilized in PBS0.2%-TritonX-100" (line 322)?

110) Please replace "TrisBase-100mM, EDTA-10mM, Tween20-0.05%; pH9" with "100 mM Tris, 10 mM EDTA, 0.05% Tween20, pH 9" (line 323).

111) Please change "99°C" to "99 °C" (line 324).

112) It is not clear what was the concentration of TritonX-100 in the washing and blocking buffer in "After washing (PBS-0.1%, Tween20-0.05%, TritonX-100), and blocking (PBS-0.2%, milk-5%, FBS-0.1%, 325 Triton X-100) for 10min, the slides were incubated overnight at 4°C with a 1:50 anti γ-H2AX mouse monoclonal antibody (Millipore, Burlington, VT, USA) and then with a 1:500 anti-mouse IgG-Alexa Fluor 488 at 37°C for 2h (ThermoFischer Scientific)" (line 324)?

113) Please provide city and state for Thermo Fisher Scientific headquarters in "After washing (PBS-0.1%, Tween20-0.05%, TritonX-100), and blocking (PBS-0.2%, milk-5%, FBS-0.1%, 325 Triton X-100) for 10min, the slides were incubated overnight at 4°C with a 1:50 anti γ-H2AX mouse monoclonal antibody (Millipore, Burlington, VT, USA) and then with a 1:500 anti-mouse IgG-Alexa Fluor 488 at 37°C for 2h (ThermoFischer Scientific)" (line 324).

114) Please replace "PBS-0.1%, Tween20-0.05%" with "0.1% PBS, 0.05% Tween20" (line 325).

115) Please change "PBS-0.2%, milk-5%, FBS-0.1%" to "0.2% PBS, 5% milk, 0.1% FBS" (line 325).

116) Please replace "10min" with "10 min" (lines 326, 329).

117) Please change "overnight at 4°C with a 1:50 anti γ-H2AX mouse monoclonal antibody (Millipore, Burlington, VT, USA)" to "with 1:50 anti γ-H2AX mouse monoclonal antibody (Millipore, Burlington, VT, USA) overnight at 4 °C" (line 326).

118) Please replace "ThermoFischer" with "Thermo Fisher" (line 328).

119) It is not exactly clear what the authors mean by "PBS-0.1%, Tween20" in "After washing in PBS-0.1%, Tween20, and PBS alone, slides were incubated 10min in 1μg.mL-1 DAPI and finally mounted with Fluoromount® (Merck)" (line 328)?

120) Please provide city and state for Merck headquarters in "After washing in PBS-0.1%, Tween20, and PBS alone, slides were incubated 10min in 1μg.mL-1 DAPI and finally mounted with Fluoromount® (Merck)" (line 328).

121) Please change "10min in 1μg.mL-1 329 DAPI and finally" to "in 1μg.mL-1 DAPI for 10 min and" (line 329).

122) Please replace "P<0.05 (*P<0.05, **P<0.01 and ***P<0.005)" with "P < 0.05 (* P <0.05, ** P < 0.01, *** P < 0.005)" (line 336).

123) Please change "(EasyRoc 1.3.1)" to "using EasyRoc 1.3.1" (line 339).

124) Please replace "of treatment" with "to treatment" (line 345).

125) Please change "pATM" to "pATM," (line 345).

126) Please change "status of patients" to "patient status" or "patient outcome" (line 346).

127) Please replace "Methodology" with "Methodology:" (line 353).

128) Please change "analysis" to "analysis:" (line 354).

129) Please replace "funding" with "funding:" (line 355).

130) Please change "preparation" to "preparation:" (line 355).

131) Please replace "editing" with "editing:" (line 356).

132) "AG" appears twice in "Writing—review and editing AL, PP, AG, AG, ASW, and CRL" (line 356). Please fix.

133) "This study (NCT02714920) was conducted in compliance with the French legislation and was approved by the local independent ethics committee (CPP Sud-Est II, Bron, France) on December 16th, 2015 , number 2015-47" (line 361) repeats from "This study (NCT02714920) was conducted in compliance with the French legislation and was approved by the local independent ethics committee (CPP Sud-Est II, Bron, France) on December 16, 2015" (line 254). Please fix. 

134) Please format "th" in "16th" using superscript (line 363).

135) Please change "2015 ," to "2015," (line 363).

136) Please replace "In The" with "The" (line 367).

137) Please change "and/or" to "and" or "or" (line 367).

138) Please replace "Dr" with "Dr." (line 369).

139) Please change "Mrs" to "Mrs." (line 370).

140) Please provide affiliation for "Dr Fabien Subtil" (line 369), "the clinical research associates’ team" (line 370), "Mrs Bénédicte Poumaroux" (line 370), and "Stéphanie Vicente" (line 371).

141) Please replace "English-edited" with "edited" (line 372).

142) Please also provide city and state for OLE-English (line 372) and LXRepair Company (line 373) office.

Author Response

We would like to thank the reviewers for their evaluation and pertinent comments, which have helped us to improve the quality of our manuscript. Please, find below the answers provided in response to all of their comments.

POINT-BY-POINT REPONSES TO THE REVIEWERS

REVIEWER 2

We greatly thank the reviewer for his/her careful proofreading which greatly improves the manuscript and for highlighting that “The manuscript is extraordinarily well prepared and the conclusions are fully supported by the acquired data”, and also that our work “has significantly pushed forward our ability to clinically diagnose HNSCC”. All issues were considered and modifications, highlighted in blue, were done as required. We hope that the corrections will meet his/her expectations.

Minor points :

1) Please provide an example of how was the area under the curve analysis presented in Table 2 performed in a new figure panel or a new figure. This could also be a new supplementary figure.

As recommended, a supplementary figure (Supplementary Figure 1) showing the ROC curves has been inserted at the end of the manuscript and an explanation was added in the corresponding section.

2) Please incorporate Supplementary Figure 1 as Figure 5 into the manuscript.

The modification was made.

3) The term "oral squamous cell carcinoma" appears only in the main title but nowhere in the text. Please fix. 

Thank you for this comment. The word oral squamous cell carcinoma has been added several times in the text in order to support the title.

4) It is not clear whether "Gersende ALPHONSE" is a corresponding author as the asterisk seems to be missing?

Mrs. Gersende ALPHONSE is one of the corresponding author and the missing asterisk was added after her lastname.

5) It is not clear what is the difference between "Univ Lyon" and "Lyon 1 University" as part of number 1 affiliation? Please provide unambiguous and full name for The University of Lyon.

The University of Lyon is divided into 3 Universities. We are part of the Lyon 1 University. The affiliation used is the official nomenclature and it is not possible to modify it.

6) Please translate "Hospices Civils de Lyon" (lines 12, 14), "Cancéropôle" (line 357), "Conseil Régional" (line 358), "Conseil Général du" (line 358), "Ligue de" (line 358), "Université de" (line 359), "Investissements d'Avenir" (line 359) into English language.

The corrections were made when it was possible since for some of them, this is the official term requested by our administration.

7) Please change "Lyon France" to "Lyon, France" (line 12).

The correction was made.

8) Please replace "UMR7516 “CHIMERE”, University of Picardie Jules Verne, Department of Biochemistry" with "Department of Biochemistry, UMR7516 “CHIMERE”, University of Picardie Jules Verne" (line 17).

For more clarity, a correction, different from the one proposed, was made. The Department of Biochemistry does not depend on the university of Picardie but from the hospital of Amiens.  

9) Please change "head-and-neck" to "head and neck" (lines 26, 46).

The corrections were made.

10) Please replace "then clearly" with "clearly" (line 28).

The correction was made.

11) Please change "ex vivo, could" to "ex vivo could" (line 29).

The correction was made.

12) Please replace "days" with "d" (lines 32, 101, 103, 104, 208, 296).

The corrections were made except in the abstract where the guidelines of the journal recommend to avoid abbreviations.

13) Please change "with" to "with the" (line 32).

The sentence was deleted

14) Please replace "prognosis" with "prognostic" (line 34).

Prognosis was changed for predictive, which is more appropriate.

5) Please change "differentiation" to "stratification" (line 35).

The correction was made.

16) Please change "24h" to "24 h" (lines 36, 158, 162, 170, 178 2x, 225, 282).

The corrections were made.

17) Please replace "4Gy" with "4 Gy" (lines 36, 121, 136, 147, 158, 159, 178 2x, 179, 195, 225, 280).

The corrections were made.

18) Please change "0Gy" to "0 Gy" (lines 37, 136, 159, 178 2x, 179, 195).

The corrections were made.

19) Please replace "improves" with "improved" (line 38).

The correction was made.

20) Please change "Damage Response" to "damage response" (line 41).

The correction was made.

21) Please replace "Head-and-Neck" with "Head and neck" (line 45).

The correction was made.

22) Please change "[5-FU]" to "(5-FU)," (line 47).

The correction was made.

23) Please replace "assigned to" with "assigned with" (line 51).

The correction was made.

24) Please change "due to less availability of data, are not yet transferred to the clinic" to "are not yet transferred to the clinic due to the lack of available data" (line 60).

The correction was made.

25) Please replace "[10,12,13] but" with "[10,12,13], but" (line 64).

The sentence was modified and "[10,12,13] but" was changed by "[10,12,13]."

26) Please change "and" to "and," (lines 68, 89, 217, 252, 353, 355).

The corrections were made.

27) Please change "ataxia-telangiectasia-mutated" to "ataxia-telangiectasia mutated" (line 69).

The correction was made.

28) "The expression of both pATM and γ-H2AX thus plays a critical role in controlling DNA repair [17] and their expression could be used as prognosis biomarkers of patient survival. In patients with nasopharyngeal cancer receiving chemoradiotherapy or with glioblastoma multiforme, high basal ATM protein expression is associated with poor overall survival (OS)" (line 72) is too long. Please split into two sentences.

The correction was made.

29) Please replace "in vivo after" with "after" (line 79).

The correction was made.

30) Please change "tumoral" to "tumour" (line 83).

The sentence was completely changed.

31) Please provide more space between line 95 and Table 1.

The correction was made.

32) From Table 1 is not clear how many non-responder patients died?

The number of patients with partial response or dead was added in the table 1.

33) Please replace "No Responders" with "Non-responders" in Table 1.

The correction was made.

34) Please change "Fig.1A" to "Fig. 1A" (line 99).

The correction was made.

35) Please replace "present" with "represent" (line 100).

The correction was made.

36) Please replace "6.5±11.4%" with "6.5 ± 11.4%" (line 101).

The correction was made.

37) Please change "Fig.1C" to "Fig. 1C" (line 102).

The correction was made.

38) Please replace "30min" with "30 min" (lines 102, 296, 324).

39) Please change "3.7±2.4%" to "3.7 ± 2.4% (line 103).

The correction was made.

40) Please replace "27.9±7.7%" with "27.9 ± 7.7%" (line 103).

The correction was made.

41) Please indicate statistical significance using asterisk(s) in Figures 1B,D, 2C, 3C.

The statistical significance was added to the figures.

42) Please define abbreviation for "SD" (line 108), "ROC" (line 188), "CI" (Table 2), "IHC" (line 354), "IF" (line 354).

The corrections were made. The abbreviations were defined at their first use. IF and IHC abbreviations are now explained in the Materials and Methods section.

43) Please change "±S D" to "±SD" (line 111).

The correction was made.

44) Please replace "(Fig.2). Fig.2A" with "(Fig. 2). Fig. 2A" (line 118).

The correction was made.

45) Please change "Fig.2B displays the means" to "Fig. 2B displays the mean" (line 119).

The correction was made.

46) Please replace "units (AI)" with "units" (line 120).

The correction was made.

47) Please replace "2256±704" with "2256 ± 704" (line 121).

The correction was made.

48) Please change "(3832±1875) (P<0.05)" to "(3832 ± 1875) (P < 0.05)" (line 121).

The correction was made.

49) Please replace "Fig.2C" with "Fig. 2C" (line 124).

The correction was made.

50) Please change "AUROC" to "area under the receiver-operating characteristic curve (AUROC)" (line 126) and "area under the receiver-operating characteristic curve (AUROC)" to "AUROC" (line 339).

The correction was made.

51) Please replace "AUC=0.789" with "AUC = 0.789" (line 130).

The sentence was changed.

52) Please increase size of the scale bar "25 μm" in both panels of Figure 2A.

The correction was made.

53) Please change "(Arbitrary Unit)" to "(Arbitrary Units)" in the y-axis title of Figure 2B.

The correction was made

54) Please change "Mean±SD" to "Mean ± SD" (line 135).

The correction was made.

55) Please replace "*P<0.05" with "* P < 0.05" (line 138).

The correction was made.

56) Please change "Figs" to "Figs." (line 144).

The correction was made.

57) Please replace "Fig.3" with "Fig. 3" (line 145).

The correction was made.

58) Please change "34.8±12.4% vs 13.1±12.6% respectively (P<0.005)" to "34.8 ± 12.4% vs 13.1 ± 12.6%, respectively, P < 0.005" (line 146).

The correction was made.

59) Please replace "Fig.3B" with "Fig. 3B" (line 147).

The correction was made.

60) Please change "39.5±15.1 %" to "39.5 ± 15.1%" (line 148).

The correction was made.

61) Please replace "(13.1±12.6%) (P<0.01)" with "(13.1 ± 12.6%) (P < 0.01)" (line 149).

The correction was made.

62) Please change "4Gy/0Gy" to "4 Gy/0 Gy" (lines 149, 169, 189, 195, 238).

The corrections were made.

63) Please change "Fig.3C" to "Fig. 3C" (line 151).

The correction was made.

64) From "Percentage±SD of pATM positive cells 24h after 0 or 4Gy irradiation for responder and non-responder patients" (line 158) is not clear whether the percentages were average?

The “Percentage ± SD” of pATM positive cells 24 h after 0 or 4 Gy irradiation for responder and non-responder patients was replace by “Average percentage ± SD” of pATM positive cells 24 h after 0 or 4 Gy irradiation for responder and non-responder patients

65) Please replace "Percentage±SD" with "Percentage ± SD" (line 158).

The correction was made.

66) Please change "*P<0.05; **P<0.01;***P<0.001" to "* P < 0.05, ** P < 0.01, *** P < 0.001" (line 161).

The corrections were made.

67) Please provide more space between lines 161 and 162.

The correction was made.

68) Please replace "Fig.4A" with "Fig. 4A" (line 162).

The correction was made.

69) Please change "Fig.4B" to "Fig. 4B" (line 163).

The correction was made.

70) Please change "number±SD" to "number ± SD" (line 164).

The correction was made.

71) Please replace "P<0.01" with "P < 0.01" (line 166).

The correction was made.

72) Please change "3.2±.5" to "3.2 ± 0.5" (line 166).

The correction was made.

73) Please replace "0.8±0.7" with "0.8 ± 0.7" (line 166).

The correction was made.

74) Please change "(2.5±2.1) and non-responder patients (3.9±2.3). Fig.4C" to "(2.5 ± 2.1) and non-responder patients (3.9 ± 2.3). Fig. 4C" (line 168).

The corrections were made.

75) Please replace "P<0.05" with "P < 0.05" (line 170).

The correction was made.

76) It is not clear what the authors refer to by "both were statistically significant" in "The threshold for classifying patients has been determined at 1.4 γ-H2AX foci per cell without irradiation and at 1.6 for the ratio, because both were statistically significant" (line 171)?

The sentence was changed by: “The threshold for classifying patients has been determined at 1.4 g-H2AX foci for non-irradiated cells. Moreover, in contrast to CD44 and pATM markers, the g-H2AX ratio 4 Gy/0 Gy was significantly different between responders and non-responders. The threshold ratio was therefore calculated and set at 1.6.”

77) Please change "foci±SD" to "foci ± SD" (line 178).

The correction was made.

78) Please replace "*P<0.05, **P<0.01" with "* P < 0.05, ** P < 0.01" (line 181).

The corrections were made.

79) It is not clear why the combination of ratio (γ-H2AX) and non-ratio (CD44) parameter "γ-H2AX Ratio, CD44" is suggested as a prognostic marker in Table 2? 

These data have been removed from Table 2

80) Please change "Ratio" to "ratio" in Table 2 10x.

The corrections were made.

81) It is not clear what the authors mean by "unique treatment" in "Here, in contrast to others who focused on a unique treatment, we showed that ex-vivo culture could be used for the determination of biomarkers prognosis of the response to both chemo- and radiotherapy in patients with HNSCC" (line 210)?

The sentence was replaced by: “Here, in contrast to others, who focused on a single treatment (radiotherapy or chemotherapy), we showed that ex vivo culture could be used for the determination of biomarkers predictive of the response to both chemo- and radio-therapy in patients with oral squamous cell carcinoma or oropharynx tumour.”

82) Please replace "others" with "others," (line 210).

The correction was made.

83) Please change "biomarkers" to "biomarker" (line 212).

The correction was made.

84) It is not clear what the authors mean by "both treatments" in "Our results highlighted that, before any treatment, high pATM expression is correlated with a poor response to both treatments" (line 224)?

The sentence was replaced by: “Our results highlighted that high pATM expression is correlated with a poor response to both chemo- or radiotherapy treatments.”

85) Please replace "that, before any treatment, high" with "that high" (line 224).

The correction was made.

86) Please change "24h after a 4Gy irradiation the percentage of pATM remains high" to "the percentage of pATM remains high 24h after a 4Gy irradiation" (line 225).

The correction was made.

87) Please replace "irradiation" with "irradiation," (line 226).

The correction was made.

88) Please change "radiation-response" to "radiation response" (line 230).

The correction was made.

89) "The majority of studies that focused on the role of γ-H2AX foci as a biomarker to predict tumour radiosensitivity [37,38] have not been performed in a clinical study but were conducted in vitro or in HNSCC tumour xenograft models" (line 234) is not semantically correct because it is hard to conceive that a "study" could be "performed in a clinical study".

The sentence was replaced as follows: “The majority of published works that focused on the role of γ-H2AX foci as a biomarker to predict tumour radiosensitivity [37,38] were conducted in vitro or in HNSCC tumour xenograft models and not in a clinical study.

90) It is not clear what the authors mean by "Adding irradiation to the ex-vivo culture" in "Adding irradiation to the ex-vivo culture allows to increase the AUC, and therefore the sensitivity of the test" (line 239)?

The sentence was changed as follows: “By irradiating the biopsy sections maintained in ex vivo culture and calculating the 4 Gy/0 Gy ratio, the AUC, and thus the sensitivity of the test, were increased”.

91) Please replace "in-vivo irradiated tumour" with "tumours irradiated in vivo" (line 242).

The correction was made.

92) Please change "Head-and-Neck" to "Head and Neck" (line 248).

The correction was made.

93) Please replace "followed-up" with "followed" (line 253).

Followed-up was replaced by monitored.

94) Please replace "minor" with "minors" (line 257).

The correction was made.

95) Please change "Fig.1" to "Fig. 1" (line 265).

The correction was made and Fig.1 was replaced by Fig. 5.

96) Please replace "tumor" with "tumour" and "Tumor" with "Tumour" 2x in Supplementary Figure 1.

The correction was made.

97) Please replace "150mL" with "150 mL" (line 274).

The correction was made.

98) It is not exactly clear what the authors refer to as "DMEM-10% FBS" in "Tumor slices of 300μm were collected and placed in 6-well plates containing DMEM-10% FBS" (line 275).

To clarify this point, the sentence was modified as follows: “Tumour slices of 300 μm were collected and placed in 6-well plates containing a mix of DMEM and 10 % FBS.”

99) Please change "Tumor" to "Tumour" (line 275).

The correction was made.

100) Please replace "300μm" with "300 μm" (line 275).

The correction was made.

101) Please change "37°C" to "37 °C" (lines 276, 307, 328).

The corrections were made.

102) Please replace "X-RAD320 Irradiator" with "X-Rad320 irradiator" (line 280).

The correction was made.

103) Please change "1h" to "1 h" (line 285).

The correction was made.

104) Please specify the microtome used in "Sections of 5μm thickness were finally cut with a Microtome (Leica) and mounted on SuperFrost Gold Slides (Fisher Scientific, Illkirch, France)" (line 286).

The correction was made.

105) Please replace "5μm" with "5 μm" (line 286).

The correction was made.

106) Please change "n=4" to "n = 4" (line 296).

The correction was made.

107) From "However, tissue sections were incubated overnight at 4°C with a 1:100 CD44 rabbit monoclonal antibody (E7K2Y,Cell Signaling, Danvers, MA, USA) in EnVision FLEX Antibody Diluent or 2h at 37°C with 1:70 pATM (phospho S1981, Abcam, Cambridge, UK)" (line 305) is not clear whether tissue sections were incubated with the pATM (phospho S1981, Abcam, Cambridge, UK) antibody in EnVision FLEX Antibody Diluent?

The sentence was modified by : “However, tissue sections were incubated with 1:100 CD44 rabbit monoclonal antibody (E7K2Y, Cell Signaling, Danvers, MA, USA) in EnVision FLEX antibody diluent overnight at 4 °C, or with 1:70 pATM rabbit monoclonal antibody (phospho S1981, Abcam, Cambridge, UK) 2 h at 37 °C in TBS, 1 % BSA.”

108) Please replace "overnight at 4°C with a 1:100 CD44 rabbit monoclonal antibody (E7K2Y,Cell Signaling, Danvers, MA, USA) in EnVision FLEX Antibody Diluent or 2h at 37°C with 1:70 pATM (phospho S1981, Abcam, Cambridge, UK)" with "with 1:100 CD44 rabbit monoclonal antibody (E7K2Y, Cell Signaling, Danvers, MA, USA) in EnVision FLEX antibody diluent overnight at 4 °C or with 1:70 pATM (phospho S1981, Abcam, Cambridge, UK) 2 h at 37 °C" (line 305).

The corrections were made as suggested by the reviewer.

109) It is not exactly clear what the authors mean by "PBS0.2%-TritonX-100" in "The slides were then immersed in unmasking buffer (TrisBase-100mM, EDTA-10mM, Tween20-0.05%; pH9) and incubated 30min at 99°C before being permeabilized in PBS0.2%-TritonX-100" (line 322)?

To clarify this point, the sentence was modified as follows: “The slides were then incubated in unmasking buffer (100 mM TrisBase, 10 mM EDTA, 0.05 % Tween 20, pH9) for 30 min at 99 °C, before being permeabilized in PBS, 0.2 % TritonX-100”.

110) Please replace "TrisBase-100mM, EDTA-10mM, Tween20-0.05%; pH9" with "100 mM Tris, 10 mM EDTA, 0.05% Tween20, pH 9" (line 323).

The corrections were made.

111) Please change "99°C" to "99 °C" (line 324).

The correction was made.

112) It is not clear what was the concentration of TritonX-100 in the washing and blocking buffer in "After washing (PBS-0.1%, Tween20-0.05%, TritonX-100), and blocking (PBS-0.2%, milk-5%, FBS-0.1%, 325 Triton X-100) for 10min, the slides were incubated overnight at 4°C with a 1:50 anti γ-H2AX mouse monoclonal antibody (Millipore, Burlington, VT, USA) and then with a 1:500 anti-mouse IgG-Alexa Fluor 488 at 37°C for 2h (ThermoFischer Scientific)" (line 324)?

The sentence was replaced by: “After washing (PBS, 0.1 % Tween 20, 0.05 % TritonX-100), and blocking (PBS, 0.2 % milk, 5 % FBS, 0.1 % Triton X-100) for 10 min, the slides were incubated with a 1:50 γ-H2AX mouse monoclonal antibody (Millipore, Burlington, VT, USA) overnight at 4 °C, and then with a 1:500 anti-mouse IgG-Alexa Fluor 488 at 37 °C for 2 h (Thermo Fischer Scientific, Waltham, MA, USA)

113) Please provide city and state for Thermo Fisher Scientific headquarters in "After washing (PBS-0.1%, Tween20-0.05%, TritonX-100), and blocking (PBS-0.2%, milk-5%, FBS-0.1%, 325 Triton X-100) for 10min, the slides were incubated overnight at 4°C with a 1:50 anti γ-H2AX mouse monoclonal antibody (Millipore, Burlington, VT, USA) and then with a 1:500 anti-mouse IgG-Alexa Fluor 488 at 37°C for 2h (ThermoFischer Scientific)" (line 324).

The corrections were made.

114) Please replace "PBS-0.1%, Tween20-0.05%" with "0.1% PBS, 0.05% Tween20" (line 325).

The corrections were made.

115) Please change "PBS-0.2%, milk-5%, FBS-0.1%" to "0.2% PBS, 5% milk, 0.1% FBS" (line 325).

The corrections were made.

116) Please replace "10min" with "10 min" (lines 326, 329).

The correction was made.

117) Please change "overnight at 4°C with a 1:50 anti γ-H2AX mouse monoclonal antibody (Millipore, Burlington, VT, USA)" to "with 1:50 anti γ-H2AX mouse monoclonal antibody (Millipore, Burlington, VT, USA) overnight at 4 °C" (line 326).

The correction was made.

118) Please replace "ThermoFischer" with "Thermo Fisher" (line 328).

The correction was made.

119) It is not exactly clear what the authors mean by "PBS-0.1%, Tween20" in "After washing in PBS-0.1%, Tween20, and PBS alone, slides were incubated 10min in 1μg.mL-1 DAPI and finally mounted with Fluoromount® (Merck)" (line 328)?

The sentence was changed by: “After 3 washing in PBS, 0.1 % Tween 20, and 2 in PBS alone, slides were incubated in 1 µg.mL-1 DAPI for 10 min, and finally mounted with Fluoromount® (Merck, Darmstadt, Germany).”

120) Please provide city and state for Merck headquarters in "After washing in PBS-0.1%, Tween20, and PBS alone, slides were incubated 10min in 1μg.mL-1 DAPI and finally mounted with Fluoromount® (Merck)" (line 328).

The correction was made.

121) Please change "10min in 1μg.mL-1 329 DAPI and finally" to "in 1μg.mL-1 DAPI for 10 min and" (line 329).

The correction was made.

122) Please replace "P<0.05 (*P<0.05, **P<0.01 and ***P<0.005)" with "P < 0.05 (* P <0.05, ** P < 0.01, *** P < 0.005)" (line 336).

The corrections were made.

123) Please change "(EasyRoc 1.3.1)" to "using EasyRoc 1.3.1" (line 339).

The correction was made.

124) Please replace "of treatment" with "to treatment" (line 345).

The correction was made.

125) Please change "pATM" to "pATM," (line 345).

The correction was made.

126) Please change "status of patients" to "patient status" or "patient outcome" (line 346).

The correction was made.

127) Please replace "Methodology" with "Methodology:" (line 353).

The correction was made.

128) Please change "analysis" to "analysis:" (line 354).

The correction was made.

129) Please replace "funding" with "funding:" (line 355).

The correction was made.

130) Please change "preparation" to "preparation:" (line 355).

The correction was made.

131) Please replace "editing" with "editing:" (line 356).

The correction was made.

132) "AG" appears twice in "Writing—review and editing AL, PP, AG, AG, ASW, and CRL" (line 356). Please fix.

This is not a mistake, AG represents both Arnaud Gauthier and Antoine Galmiche. The abbreviations were detailed in the affiliation part, and Antoine Galmiche is now presented as AGA.

133) "This study (NCT02714920) was conducted in compliance with the French legislation and was approved by the local independent ethics committee (CPP Sud-Est II, Bron, France) on December 16th, 2015 , number 2015-47" (line 361) repeats from "This study (NCT02714920) was conducted in compliance with the French legislation and was approved by the local independent ethics committee (CPP Sud-Est II, Bron, France) on December 16, 2015" (line 254). Please fix. 

According to the journal's guidelines, it is mandatory to have an “Institutional Review Board Statement” section. To this end, only the Materials and Methods section has been changed.

134) Please format "th" in "16th" using superscript (line 363).

The correction was made.

135) Please change "2015 ," to "2015," (line 363).

The correction was made.

136) Please replace "In The" with "The" (line 367).

The correction was made.

137) Please change "and/or" to "and" or "or" (line 367).

The correction was made.

138) Please replace "Dr" with "Dr." (line 369).

The correction was made.

139) Please change "Mrs" to "Mrs." (line 370).

The correction was made.

140) Please provide affiliation for "Dr Fabien Subtil" (line 369), "the clinical research associates’ team" (line 370), "Mrs Bénédicte Poumaroux" (line 370), and "Stéphanie Vicente" (line 371).

The affiliations were added.

141) Please replace "English-edited" with "edited" (line 372).

The correction was made.

142) Please also provide city and state for OLE-English (line 372) and LXRepair Company (line 373) office.

The corrections were made.

Round 2

Reviewer 1 Report

The authors have properly addressed all my comments.